



# A Comparative Study to Reveal the Influence of Typhoons on the Transport, Production and Accumulation of O₃ in the Pearl River Delta, China

Kun Qu[1,2], Xuesong Wang[1,2], Yu Yan[1,2], Jin Shen[3], Teng Xiao[1,2], Huabin Dong[1,2], Limin Zeng[1,2], and Yuanhang Zhang[1,2,4,5]

[1]State Key Joint Laboratory of Environmental Simulation and Pollution Control, College of Environmental Sciences and Engineering, Peking University, Beijing 100871, China
[2]International Joint Laboratory for Regional Pollution Control, Ministry of Education, Beijing, 100816, China
[3]State Key Laboratory of Regional Air Quality Monitoring, Guangdong Key Laboratory of Secondary Air Pollution Research, Guangdong Environmental Monitoring Center, Guangzhou 510308, China
[4]Beijing Innovation Center for Engineering Science and Advanced Technology, Peking University, Beijing 100871, China
[5]CAS Center for Excellence in Regional Atmospheric Environment, Chinese Academy of Sciences, Xiamen 361021, China

*Correspondence to*: Xuesong Wang (xswang@pku.edu.cn) and Yuanhang Zhang (yhzhang@pku.edu.cn)

**Abstract.** The Pearl River Delta (PRD) region in South China is faced with severe ambient O₃ pollution in autumn and summer, which mostly coincides with the occurrence of typhoons above the Northwest Pacific. With increasingly severe O₃ pollution in the PRD under the influence of typhoons, it is necessary to gain a comprehensive understanding of the impact of typhoons on O₃ transport, production and accumulation for efficient O₃ reduction. In this study, we analysed the general influence of typhoons on O₃ pollution in the PRD via systematic comparisons of meteorological conditions, O₃ processes and sources on O₃ pollution days with and without typhoon occurrence (denoted as the typhoon-induced and no-typhoon scenarios, respectively), and also examined the differences in these influences in autumn and summer. The results show that the approach of typhoons was accompanied by higher wind speeds and strengthened downdrafts in autumn as well as the inflows of more polluted air masses in summer, suggesting favourable O₃ transport conditions in the typhoon-induced scenario in both seasons. However, the effect of typhoons on the production and accumulation of O₃ were distinct. Typhoons led to reduced cloud cover, and thus stronger solar radiation in autumn, which accelerated O₃ production, but the shorter residence time of local air masses was unfavourable for the accumulation of O₃ within the PRD. In contrast, in summer, typhoons increased cloud cover, and weakened solar radiation, thus restraining O₃ formation, but the growing residence time of local air masses favoured O₃ accumulation. The modelling results using the Community Multiscale Air Quality (CMAQ) model for the typical O₃ pollution days suggest increasing contributions from the transport processes as well as sources outside the PRD for O₃ pollution, confirming enhanced O₃ transport under typhoon influence in both seasons. The results of the process analysis in CMAQ suggest that the chemical process contributed more in autumn but less in summer in the PRD. Since O₃ production and accumulation cannot be enhanced at the same time, the proportion of O₃ contributed by emissions within the PRD was likely to decrease in both seasons. The difference in the typhoon influence on O₃ processes in autumn and summer can be attributed to the seasonal variation of the East Asian monsoon. From the "meteorology-process-source" perspective, this study revealed





the complex influence of typhoons on O$_3$ pollution in the PRD and their seasonal differences. To alleviate O$_3$ pollution under
typhoon influence, emission control is needed on a larger scale, rather than only within the PRD.

## 1 Introduction

Tropospheric ozone (O$_3$) serves as a secondary pollutant in ambient air and is detrimental for human health and crop
production (Wang et al., 2017; Liu et al., 2018; Mills et al., 2018). Ambient O$_3$ is produced from its precursors, i.e., nitrogen
oxides (NO$_x$ = NO + NO$_2$) and volatile organic compounds (VOCs), through chemical reactions in the presence of sunlight.
This O$_3$ can accumulate locally, or be transported to downwind regions. Under unfavourable meteorological conditions,
enhanced transport, production and/or accumulation of O$_3$ can all contribute to the O$_3$ pollution within a region (National
Research Council, 1991).

As the largest city cluster in South China, the Pearl River Delta (PRD) region is faced with frequent ambient O$_3$ pollution,
especially in autumn and summer (Li et al., 2014; Wang et al, 2017; Lu et al, 2018). Along with the continuous increasing of
O$_3$ levels in recent years (Li et al., 2019), O$_3$ has become the primary contributor to the deterioration of air quality in this
region (Feng et al., 2019). The occurrence of O$_3$ pollution in the PRD is predominantly related to the influence of typhoons
(or tropical cyclones) above the Northwest Pacific (Gao et al., 2018; Deng et al., 2019; Lin et al., 2019). According to Gao et
al. (2018), seven out of the nine most severe O$_3$ episodes (regional-mean maximum 8-h average O$_3$ concentrations of > 240
μg/m$^3$) during 2014–2016 coincided with the approach of typhoons. The changes in the track and intensity of typhoons may
contribute to the growing trend of O$_3$ levels recently and in future (Lam, 2018; Lam et al., 2018). Therefore, a
comprehensive understanding of the influence of typhoons on the transport, production and accumulation of O$_3$ has
important implications for efficient and strategic O$_3$ reduction in the PRD.

Analyses of typhoon-related O$_3$ episodes in the PRD have been extensively reported in previous publications. The effect of
typhoons on O$_3$ pollution is closely linked to meteorological conditions that are conducive to the transport, production and/or
accumulation of O$_3$. Stagnation caused by typhoons, characterised by low wind speeds, has been reported during many
episodes, and it promotes the accumulation of locally formed O$_3$ within the PRD (Wang et al., 1998; So and Wang, 2003;
Wang and Kwok, 2003; Huang et al., 2005; Lam et al., 2005; Jiang et al., 2008; Zhang et al., 2014; Chow et al., 2019).
Strong north or west winds were observed or simulated during several episodes, suggesting the potentially strengthened
transport of pollutants under typhoon influence (Wang et al., 2001; Yang et al., 2012; Wang et al., 2015; Wei et al., 2016).
Downdrafts on the outskirts of typhoons may promote downward O$_3$ transport and contribute to near-ground O$_3$ pollution as
well (Lam, 2018), but its appearance in the PRD has only been examined in a few studies. Cloudless conditions and strong
solar radiation enhance O$_3$ production, which is another important cause of O$_3$ pollution (Wang et al., 1998; Wang and
Kwok, 2003; Li et al., 2018; Yue et al., 2018; Chow et al., 2019). In a more direct way, several studies have utilised





chemical transport models, along with the Process Analysis (PA) tool and source apportionment (SA) methods, to quantify
and compare the contributions of various $O_3$ processes (e.g., transport and the chemical process) and sources (e.g., local
emissions, outside emissions and background) during these episodes. Based on reports by Huang et al. (2005), Lam et al.
(2005), Jiang et al. (2008), Wang et al. (2010), Li (2013), Wang et al. (2015), Wei et al. (2016) and Chen et al. (2018),
horizontal/vertical transport and chemical production may both be the main contributing process for typhoon-induced $O_3$
pollution in different parts of the PRD. The SA results revealed that emissions within the PRD contributed 40–80% of $O_3$
during typhoon-related $O_3$ episodes (Li et al., 2012; Li, 2013; Chen et al., 2015), suggesting the potentially important role of
$O_3$ accumulation for $O_3$ pollution here. However, despite massive episode-based studies, several important questions still
remain: Are $O_3$ transport, production and accumulation within the PRD all enhanced at the same time by typhoons? Do both
$O_3$ pollution seasons (autumn and summer) experience similar impact of typhoons on $O_3$ pollution? More thorough
investigations are needed to answer these questions.

In this study, we present systematic comparisons between $O_3$ pollution in the typhoon-induced and no-typhoon scenarios
(definitions given in Sect. 2.2) to elucidate the influence of typhoons on $O_3$ transport, production and accumulation in the PRD
and to reveal their seasonal differences. October and July in 2014–2018 were selected as the representative months for autumn
and summer, respectively. Multiple datasets, including the ERA-Interim re-analysis, the routine monitoring datasets,
trajectories calculated by the Hysplit model and the modelling results of typical $O_3$ pollution days using the Community
Multiscale Air Quality (CMAQ) model, were used in the comparisons. A detailed introduction of these datasets is presented
in Sect. 2. The comparisons were conducted from the perspectives of meteorological conditions (Sect. 3), $O_3$ processes and
sources (Sect. 4), and the conclusions about the influence of typhoons on the causes of ambient $O_3$ pollution in the PRD in the
two seasons are illustrated in Sect. 5.
**2 Methods**
**2.1 Datasets**
The detailed information for the datasets utilised in the comparison of meteorological conditions is presented below:
•   **Re-analysis datasets**: We mainly used the ERA-Interim re-analysis product in the analyses due to its more available

parameters and high spatial coverage (available at https://www.ecmwf.int/en/forecasts/datasets/reanalysis-datasets/era-

interim, last accessed: March 2020; Dee et al., 2011; Berrisford et al., 2011). Specifically, meteorological parameters

used in the comparisons include the following three categories: (1) near-surface parameters from the analysis fields,

including air temperature (at a height of 2 m), relative humidity (RH, at 1000 hPa), horizontal wind speeds (at a height

of 10 m; zonal and meridional wind speeds were also involved in the comparisons), and low (for the height at which

pressure/surface pressure > 0.8), medium (for the height at which 0.45 < pressure/surface pressure < 0.8), high (for the

height at which pressure/surface pressure < 0.45) and total cloud covers; (2) near-surface parameters from the forecast



fields, including plenary boundary layer (PBL) height and net surface solar radiation; and (3) upper air parameters at
multiple heights, including horizontal and vertical wind speeds, cloud water content and $O_3$ mixing ratio. The focus of
this study is $O_3$ pollution during the daytime, and therefore, only the parameters at 14:00 local time (LT) were selected
for the analyses (except for net surface solar radiation, which was averaged within 8:00–17:00 LT).
•    **Surface meteorological routine monitoring datasets**: The routine monitoring meteorological data collected at 29

national meteorological sites within the PRD (locations shown in Fig. S1a) were also used to explore the

meteorological features under the impact of typhoons. The parameters include air temperature, RH, and wind speed and

direction (also transformed to zonal and meridional wind speeds in the comparisons) at 14:00 LT.

•    **Typhoon information**: The times, locations and intensities of typhoons were provided by the Chinese Meteorological

Administration Best Track Dataset of tropical cyclones (Ying et al., 2014). The tracks of all typhoons that potentially

contributed to $O_3$ pollution in the PRD during the study period (October and July in 2014–2018) are shown in Fig. S2

and S3.

•    **$O_3$ concentrations**: Hourly $O_3$ concentration data, which were originally released by the China National Environmental

Monitoring Centre, were downloaded from http://beijingair.sinaapp.com (last accessed: Dec. 2018). Based on the

hourly data, we calculated the maximum 1-hr concentrations (MDA1) and maximum 8-hr average concentrations

(MDA8) of $O_3$ in nine municipalities in the PRD (including Guangzhou, Shenzhen, Zhuhai, Foshan, Jiangmen,

Zhaoqing, Huizhou, Dongguan and Zhongshan) to identify $O_3$ pollution days that served as samples in the comparisons.

## 2.2 Definition and classification of $O_3$ pollution days

In this study, $O_3$ pollution days were defined as the days when the MDA1 exceeds 200 μg/m$^3$ or the MDA8 exceeds 160
μg/m$^3$ for $O_3$ (both are the Grade-II thresholds of the Chinese National Ambient Air Quality Standard (NAAQS), GB 3095-
2012) in any of the nine municipalities in the PRD. According to these criteria, there were 78 and 55 $O_3$ pollution days
(given in Table S1 and S2) during October and July in 2014–2018, respectively. The information about these $O_3$ pollution
days in the two representative months is listed in Table 1 (overall) and S3 (monthly), including the numbers of days, their
proportions in the month, and the corresponding mean $O_3$ concentrations (MDA8 and MDA1, highest values among nine
municipalities in the PRD). Although there were more $O_3$ pollution days in October than in July, $O_3$ pollution days under
typhoon influence accounted for ~30% of all days in both months. Higher $O_3$ MDA1 and MDA8 values can be generally
found with the appearance of typhoons in comparison with days without typhoons, further indicating the important role of
typhoons in $O_3$ pollution in the PRD.

The differing locations of typhoons can result in the diverse effect of typhoons on $O_3$ pollution (Chow et al., 2018). To
determine the general influence of typhoons on $O_3$ pollution in the PRD, it was necessary to further select $O_3$ pollution days
coinciding with typhoons with similar directions and distances from the PRD. First, we removed five $O_3$ pollution days in
July with typhoons located to the due north or southwest of the PRD from the analyses. As is shown in Fig. 1, the remaining





days, including all $O_3$ pollution days in October and most $O_3$ pollution days in July under typhoon influence, were associated
with typhoons to the east of the PRD, which were more likely to cause $O_3$ pollution (Chow et al., 2018). After this, based on
the distances between typhoon centres and the PRD (at 14:00 LT), we classified the pollution days in each season into three
categories: close typhoon (lowest 20% of distances), typhoon (20–80% intervals of distances), and far typhoon (longest 20%
of distances)-induced days. The typhoon-induced days represent $O_3$ pollution days with general typhoon influence, and they
were compared with those without the appearance of typhoons (hereafter denoted as the no-typhoon days). It should be noted
that the distances between typhoon centres and the PRD on the typhoon-induced days were overall larger in autumn (1400–
2800 km, at 14:00 LT) than in summer (700–2000 km, at 14:00 LT), which may be the consequence of the different
characteristics of typhoon paths in the two seasons: most typhoons in autumn travel northwest initially and then turn
northward in the areas east of the Philippines (Fig. S2), whereas they are more likely to end up landing in Southeast China in
summer (Fig. S3). Since the influence of typhoons on $O_3$ pollution may be different when typhoons come close enough to
the PRD (Lam et al., 2005; Li, 2013), the close typhoon-induced days were considered to be a special scenario in the
comparisons of meteorological conditions (Sect. 3.5). Owing to the less apparent effect of typhoons over the PRD, we did
not include the far typhoon-induced days in the discussions.

**Table 1.** The numbers and proportions of $O_3$ pollution days, and $O_3$ concentrations for various scenarios.

| Parameter | October, 2014–2018 | July, 2014–2018 |
|---|---|---|
| Number (proportion) of $O_3$ pollution days | 78 (50.3%) | 55 (35.5%) |
| With typhoons | 49 (31.6%) | 45 (29.0%) |
| Typhoon-induced days | 30 (19.4%) | 24 (15.5%) |
| Close typhoon-induced days | 10 (6.5%) | 8 (5.2%) |
| Without typhoons (no-typhoon days) | 29 (18.7%) | 10 (6.5%) |
| Mean PRD-max $O_3$ MDA8 ($\mu g/m^3$) | | |
| With typhoons | 195.0 | 205.3 |
| Typhoon-induced days | 199.5 | 205.4 |
| Close typhoon-induced days | 184.6 | 225.7 |
| Without typhoons (no-typhoon days) | 189.8 | 187.8 |
| Mean PRD-max $O_3$ MDA1 ($\mu g/m^3$) | | |
| With typhoons | 230.4 | 259.8 |
| Typhoon-induced days | 235.2 | 260.0 |
| Close typhoon-induced days | 219.2 | 277.1 |
| Without typhoons (no-typhoon days) | 231.5 | 246.5 |



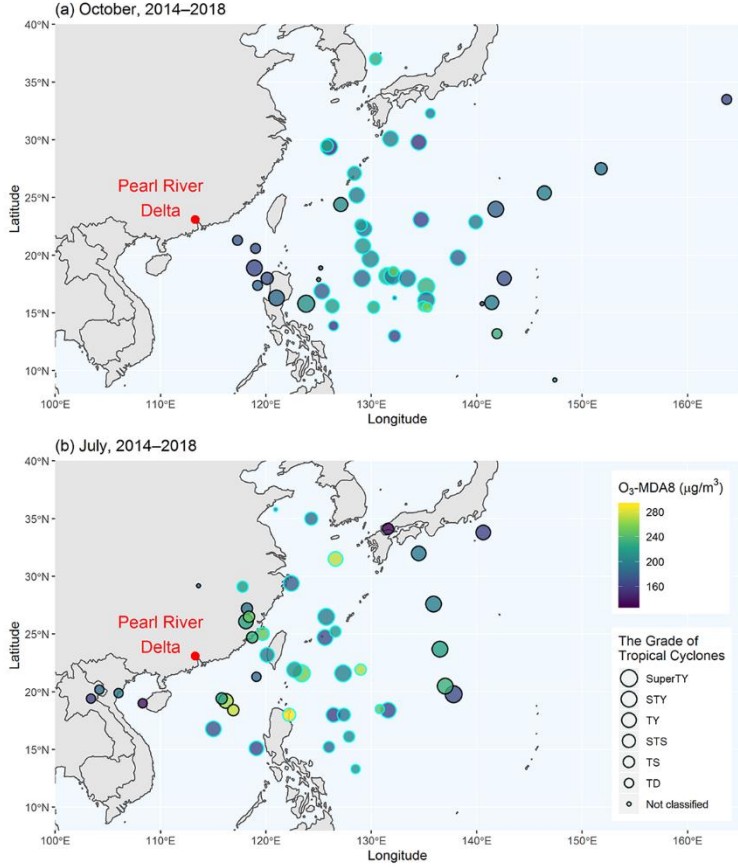


**Figure 1.** The location and intensity of typhoons at 14:00 LT on all $O_3$ pollution days with typhoons, and the corresponding $O_3$ MDA8
concentrations (maximum values in the nine municipalities of the PRD) on the same days during (a) October and (b) July in 2014–2018.
The points with cyan borders indicate the "typhoon-induced" $O_3$ pollution days used in the comparisons. The grades of tropical cyclones
(Chinese National Standard, GB/T 19201-2006) are as follows: SuperTY - super typhoon; STY - severe typhoon; TY - typhoon; STS -
severe tropical storm; TS - tropical storm; TD - tropical depression; others are grouped as "not classified".

**2.3 Calculation of the trajectories and air parcel residence time**

To explore the potential effect of cross-regional transport on $O_3$ pollution in the PRD, we applied the Hysplit model (Stein et
al., 2015) with the Global Data Assimilation System (GDAS) datasets as inputs to calculate 72-h backward trajectories reaching
the PRD at 14:00 LT for all $O_3$ pollution days. The Modiesha site (23.1 °N, 113.3 °E; Fig. S1b), which is located in the central
part of the PRD, was the endpoint of backward trajectories, with its height set as 500 m above the ground.

Air parcel residence time (APRT), discussed by Huang et al. (2019), is the average number of hours that air parcels originated
from one place stay within a pre-defined domain, and long APRTs can be used to indicate good accumulation conditions for
locally sourced pollutants. To calculate APRTs in the PRD, we designed a 21×15 point matrix that embraces the whole PRD
(Fig. S4), and forward trajectories starting from these points were also calculated using the Hysplit model. The height of all


points was set as 100 m above the ground, which is close to the height of emissions. The start times were set as 2:00, 8:00,
14:00 and 20:00 LT for all $O_3$ pollution days. Afterwards, the length of time each trajectory remained within the administration
borders of the PRD, i.e., APRT, was calculated. The APRT values of all points were averaged for each scenario and were
interpolated to obtain field results.
**2.4 CMAQ modelling: basic setups and modelling methods**
We utilised the widely used 3D chemical transport model, the CMAQ model (version 5.0.2), to investigate the effect of
typhoons on $O_3$ processes and sources. October 2015 and July 2016 featured the most severe $O_3$ pollution under typhoon
influence among all representative months of the two seasons (Table S3), and thus, they were chosen as the period in the
CMAQ modelling (because there was no severe $O_3$ pollution during the first 10 days of October 2015 and 3–5 November can
be classified as the no-typhoon $O_3$ pollution days, we adjusted the modelling period in autumn to 11 October–10 November
2015) and all $O_3$ pollution days in these two months served as representative $O_3$ pollution days under multiple scenarios. In
detail, there were four typhoon-induced $O_3$ pollution days (14–16 and 21 October 2015) and four no-typhoon $O_3$ pollution days
(28 October and 3–5 November 2015) in October 2015, whereas there were four and six typhoon-induced and no-typhoon
days in July 2016, respectively (typhoon-induced: 7–8 and 30–31 July 2016; and no-typhoon: 22–26 and 29 July 2016). The
results of the daytime (9:00–17:00 LT) $O_3$ PA and SA on the above $O_3$ pollution days were averaged for each scenario and
were used in the comparisons.

The main setups of the CMAQ modelling are presented as follows. Two-nested modelling domains with the resolutions of 36
and 12 km (denoted as d01 and d02, respectively) were set in this study (Fig. 2). Specifically, d02 covers the whole East and
Central China (EC-China), enabling us to evaluate the contribution of emissions in these areas to $O_3$ pollution in the PRD.
There were 19 vertical layers in the CMAQ modelling, with about 10 layers within the PBL (about 0–1 km in heights; Guo et
al., 2016). The Weather Research and Forecasting (WRF) model (version 3.2) provided the meteorological fields used as inputs.
SMOKE (version 2.5) and MEGAN (version 2.10) were used to process the anthropogenic and biogenic emission files,
respectively. The anthropogenic emission inventory used in this study consisted of the following three parts: (1) emissions in
the PRD, which were provided by the Guangdong Environmental Monitoring Centre; (2) emissions in other areas of mainland
China, which were extracted from the MEIC inventory (He, 2012); and (3) emissions in other countries and regions in Asia,
which were extracted from the MIX inventory (Li et al., 2017). The initial and boundary conditions of the d01 modelling were
obtained from the same-period results of the MOZART-4 global model (available at https://www.acom.ucar.edu/wrf-
chem/mozart.shtml, last accessed: Dec. 2019), and those of the d02 modelling were extracted from the d01 modelling results.
The SAPRC07 gas-phase chemistry mechanism (Carter, 2010) and the AERO6 aerosol scheme were set in the CMAQ
modelling. In addition, the simulations of the two months were both started 10 days ahead to minimise the disturbance of the
bias of the initial conditions. The modelling performances of CMAQ and WRF were determined to be acceptable based on the
comparisons between the observational and modelling series of meteorological parameters, $O_3$ MDA8, daily $NO_2$





concentrations and the mixing ratios of non-methane hydrocarbons (NMHCs) in the PRD (for details, refer to Sect. 1 of the

Supplement Information), which ensures the validity of the further analyses.

**Figure 2.** Two-nested modelling domain, noted as d01 and d02. The black boxes indicate the WRF modelling domains, and the nested
areas are the CMAQ modelling domains.

The PA tool in CMAQ was implemented to quantify the hourly contributions of $O_3$ processes (or integrated process rate, IPR),

which includes vertical/horizontal transport (convection+diffusion), chemical process (net $O_3$ production), dry deposition and

cloud process. To explore the overall effect of typhoons on $O_3$ transport and production in the region, the mean PA results

within the administration boundaries of the PRD were calculated and compared.

We used the classic Brute Force Method (BFM) to identify the contributions of emissions (including anthropogenic and

biogenic emissions) in the PRD and other regions in the d02 (mainly EC-China), as well as regions outside the d02 (the

boundary conditions of the d02) for $O_3$ pollution in the PRD (hereafter denoted as the contributions of PRD, EC-China and

BCON, or $S_{PRD}$, $S_{EC\text{-}China}$ and $S_{BCON}$, respectively). For a pollutant, the contribution of a specific emission, $E_i$, can be calculated

in two ways: (1) the difference between the modelled concentrations of the base case (all emissions involved) and the sensitivity

case where $E_i$ is zeroed out (i.e., top-down BFM); (2) the difference between two sensitivity cases where emissions expect $E_i$

and all of the emissions are zeroed out, respectively (i.e., bottom-up BFM). Owing to the non-linearity between $O_3$ and its

precursors, biases may occur between the results of two types of BFM methods (Clappier et al., 2017). Therefore, the average

of the top-down and bottom-up BFM results was treated as the quantified contributions of the concerned sources. Four

simulation cases were run in this study, including (the modelled $O_3$ concentration in each case was also marked in brackets):

- the base case ($C_{base}$);
- the PRD-cut case ($C_{PRD\_cut}$), where emissions within the PRD were zeroed out;
- the PRD-only case ($C_{PRD\_only}$), where emissions outside the PRD (within d02) were zeroed out; and
- the zero-emission case ($C_0$), where all emissions within the d02 were zeroed out.





Afterwards, the $S_{PRD}$, $S_{EC\text{-}China}$ and $S_{BCON}$ values (in concentrations) in the polluted areas of the PRD (where modelled daytime
$O_3$ concentrations $> 160\ \mu g/m^3$, the Grade-II $O_3$ MDA8 thresholds of the Chinese NAAQS) were calculated using the following
equations,

$$S_{PRD} = \tfrac{1}{2}\left[\left(C_{base} - C_{PRD\_cut}\right) + \left(C_{PRD\_only} - C_0\right)\right], \tag{R1}$$

$$S_{EC-China} = \tfrac{1}{2}\left[\left(C_{base} - C_{PRD\_only}\right) + \left(C_{PRD\_cut} - C_0\right)\right], \tag{R2}$$

$$S_{BCON} = C_0. \tag{R3}$$

The percentage forms of these values were used in the comparisons.

## 3 Comparison of meteorological conditions

### 3.1 Overview: comparison of meteorological parameters in the PRD

First, we compared near-ground meteorological parameters in the PRD on the typhoon-induced and no-typhoon $O_3$ pollution
days. The parameters from the ERA-Interim re-analysis (including the parameters of the first and second categories in Sect.
2.1) and the routine monitoring datasets (including air temperature, RH, wind speed, zonal and meridional wind speeds)
were used in the comparison. The Mann-Whitney U test was applied to determine whether the above parameters were
significantly different ($p < 0.05$) in the two types of $O_3$ pollution scenarios.

As is listed in Table 2, statistically significant differences between the typhoon-induced and no-typhoon scenarios existed for
most of the parameters, such as meridional (south-north) wind speed, cloud covers within various height ranges and net
surface solar radiation — in both seasons, these parameters were significantly different for the two scenarios. It indicates that
the causes of $O_3$ pollution may vary on typhoon-induced and no-typhoon $O_3$ pollution days. Note that air temperature, one of
the parameters most closely related to $O_3$ pollution in the PRD (Zhao et al., 2019), was not significantly different in the two
scenarios. We also found that the comparison in autumn and summer did not produce the same results: the typhoon-induced
days in autumn featured lower RH, stronger winds (especially north wind), reduced cloud cover (low, medium, high and
total) and stronger surface solar radiation, whereas in summer, these days had higher RH, weaker south winds, more cloud
cover (medium, high and total), weaker surface solar radiation and lower PBL heights. Therefore, the impact of typhoons on
$O_3$ pollution differs in the two seasons, as well. In order to reveal the impact of typhoons on $O_3$ transport, production, and
accumulation in the PRD, more detailed comparisons of the corresponding meteorological indicators are presented in the
following sections.


**Table 2.** The comparisons of meteorological parameters (all at 14:00 LT except for net surface solar radiation, which is the average value for 9:00–17:00 LT) in the PRD for the three scenarios (no-typhoon, typhoon-induced, close typhoon-induced) in two seasons (autumn, summer). RM, routine measurement; ERA, ERA-Interim re-analysis. All of the parameters are presented as "the mean value ± standard deviation". The differences between parameters in the typhoon-induced or close typhoon-induced scenarios and the corresponding typhoon-induced scenarios for the same season are given in parentheses, and "*" indicates $p < 0.05$, or statistically significant differences between these parameters when the Mann-Whitney U test is used.

| Parameters | Data Source | Autumn (October, 2014–2018) | | | Summer (July, 2014–2018) | | |
|---|---|---|---|---|---|---|---|
| | | No-typhoon | Typhoon-induced | Close Typhoon-induced | No-typhoon | Typhoon-induced | Close Typhoon-induced |
| Air Temperature (℃) | RM | 29.1 ±2.2 | 29.3 ±1.8 (+0.2) | 29.6 ±1.5 (+0.5, *) | 33.7 ±2.0 | 33.9 ±2.0 (+0.2) | 35.0 ±1.5 (+1.3, *) |
| | ERA | 29.2 ±2.1 | 29.3 ±1.6 (+0.1) | 29.6 ±1.5 (+0.4, *) | 33.4 ±1.8 | 33.5 ±1.4 (+0.1) | 34.6 ±1.4 (+1.2, *) |
| RH (%) | RM | 52.4 ±10.2 | 44.8 ±10.4 (-7.6, *) | 51.4 ±12.4 (-1.0) | 57.0 ±9.3 | 58.3 ±9.7 (+1.3) | 56.9 ±6.4 (-0.1) |
| | ERA | 54.0 ±9.8 | 48.3 ±11.2 (-5.7, *) | 52.2 ±12.4 (-1.8, *) | 62.6 ±10.8 | 66.4 ±9.4 (+3.8, *) | 62.5 ±9.4 (+0.1) |
| Wind Speed (m/s) | RM | 2.33 ±1.18 | 2.58 ±1.23 (+0.25, *) | 2.96 ±1.40 (+0.63, *) | 2.46 ±1.33 | 2.30 ±1.20 (-0.16) | 2.53 ±1.16 (+0.07) |
| | ERA | 2.39 ±1.30 | 2.54 ±0.99 (+0.15, *) | 3.53 ±1.11 (+1.14, *) | 2.41 ±0.99 | 2.18 ±1.18 (-0.23, *) | 2.61 ±1.05 (+0.20) |
| Zonal (East-West) Wind Speed (m/s) | RM | -0.83 ±1.72 | -0.59 ±1.70 (+0.24, *) | -0.13 ±1.74 (+0.70, *) | -0.41 ±2.05 | -0.03 ±1.94 (+0.38) | 0.73 ±1.98 (+1.14, *) |
| | ERA | -1.41 ±1.43 | -1.07 ±1.04 (+0.34, *) | -0.87 ±0.79 (+0.54, *) | 0.22 ±1.73 | -0.02 ±1.81 (-0.24) | 0.29 ±2.45 (+0.07) |
| Meridional (South-North) Wind Speed (m/s) | RM | -0.36 ±1.74 | -1.49 ±1.66 (-1.13, *) | -2.21 ±1.66 (-1.85, *) | 0.79 ±1.69 | 0.01 ±1.72 (-0.78, *) | -0.69 ±1.68 (-1.48, *) |
| | ERA | -0.27 ±1.82 | -1.97 ±1.16 (-1.70, *) | -3.27 ±1.29 (-3.00, *) | 1.61 ±1.09 | 0.64 ±1.58 (-0.97, *) | -0.68 ±1.19 (-2.29, *) |
| Low Cloud Cover (%) | ERA | 17.2 ±22.7 | 4.2 ±11.9 (-13.0, *) | 15.5 ±23.9 (-1.7, *) | 8.7 ±9.4 | 7.1 ±9.5 (-1.6, *) | 5.2 ±5.0 (-3.5, *) |
| Medium Cloud Cover (%) | ERA | 22.2 ±26.5 | 10.4 ±19.7 (-11.8, *) | 9.5 ±14.5 (-12.7, *) | 8.7 ±11.1 | 15.4 ±15.1 (+6.7, *) | 21.5 ±15.5 (+12.8, *) |
| High Cloud Cover (%) | ERA | 12.1 ±23.1 | 7.2 ±16.3 (-4.9, *) | 34.6 ±35.6 (+22.5, *) | 32.2 ±30.0 | 44.9 ±29.3 (+12.7, *) | 51.0 ±34.2 (+18.8, *) |
| Total Cloud Cover (%) | ERA | 43.5 ±32.3 | 20.5 ±25.7 (-23.0, *) | 51.9 ±33.1 (+8.4, *) | 43.7 ±26.7 | 58.3 ±22.7 (+14.6, *) | 67.5 ±21.0 (+23.7, *) |
| Net Surface Solar Radiation (W/m²) | ERA | 456.9 ±78.4 | 516.6 ±66.7 (+59.7, *) | 516.5 ±62.8 (+59.6, *) | 560.3 ±93.1 | 523.2 ±74.4 (-37.1, *) | 541.9 ±54.0 (-18.4, *) |
| PBL Height (m) | ERA | 1471 ±315 | 1473 ±348 (+2) | 1349 ±227 (-122, *) | 1268 ±383 | 1037 ±289 (-231, *) | 1196 ±300 (-72, *) |

## 3.2 O₃ transport conditions: comparison of wind speeds, backward trajectories and vertical air motions

The higher wind speeds and/or O₃ levels in the transported air masses are, the more likely O₃ transport plays an increasingly important role in O₃ pollution. In the PRD, O₃ levels are closely linked to the type of air masses influencing the region, which can be identified based on backward trajectories. According to Zheng et al. (2010), there are generally three types of air masses



that are transported into the PRD along different paths and contribute to O₃ pollution here, namely, the continental, coastal and
marine air masses (Fig. 3a). The continental and coastal air masses can bring O₃ from EC-China to the PRD, and thus, they
are typically recognised as being polluted and contributing to relatively high O₃ levels in the PRD. In contrast, the marine air
masses, originated from the South China Sea, are much cleaner. In this section, we studied the influence of typhoons on O₃
transport by comparing wind speeds and 72-h backward trajectories in various scenarios.

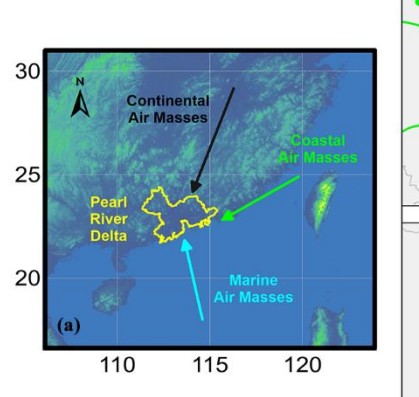
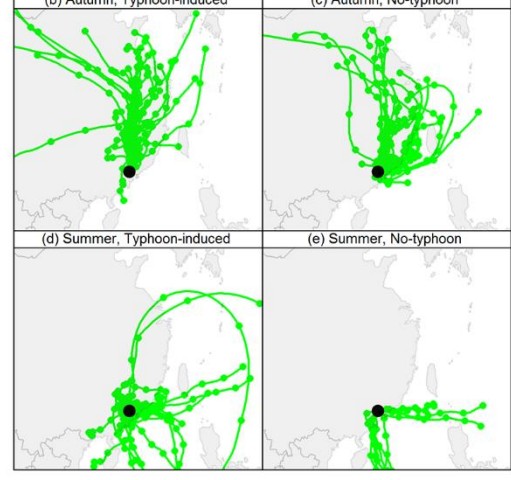


**Figure 3.** (a) Three O₃ transport paths towards the PRD. (b–e) Backward trajectories at 14:00 LT for the four scenarios: (b) autumn,
typhoon-induced; (c) autumn, no-typhoon; (d) summer, typhoon-induced; and (e) summer, no-typhoon. The black dots indicate the end
point of all trajectories, i.e., where the Modiesha site in the central PRD is located.
As is displayed in Fig. 3b–c, we identified the influence of continental air masses on the typhoon-induced O₃ pollution days
in autumn, as well as mixed contributions from the continental and coastal air masses on the corresponding no-typhoon days.
However, for the former scenario, significantly increased wind speeds (Table 2) ensure more favourable conditions for the
transport of O₃. In summer, the three types of air masses may all have affected O₃ pollution in the typhoon-induced scenario,
while only the marine air masses influenced the PRD in the no-typhoon scenario (Fig. 3d–e). Since wind speeds did not vary
significantly (Table 2), the inflows of much more polluted air masses resulted in that typhoons also tended to increase the
contribution of transport to O₃ pollution in the PRD in summer. In addition, the influence of different air masses was also
accompanied with variations in the prevailing winds in the PRD, that is, north winds and easterlies in the typhoon-induced and
no-typhoon scenario in autumn, respectively, and southwest winds in the no-typhoon scenario in summer (indicated by wind
roses in Fig. S5). For the typhoon-induced scenario in summer, the dominate wind direction is hard to determine. These
variations in the local wind fields potentially result in the different spatial distribution of O₃ concentrations in various scenarios.

Downdrafts are typically considered to be an important cause of typhoon-induced O₃ pollution (Lam, 2018), but in which
scenarios downdrafts influence the PRD remains unclear. Thus, we explored the overall features of vertical air motions from



the surface layer to the tropopause in four scenarios, and the ERA-Interim reanalysis dataset (including the parameters of the
third category in Sect. 2.1) was utilised in the comparisons. The contours in Fig. 4 show the cross section of vertical wind
speed from 26.0 °N to 20.0 °N along the 113.2 °E longitude line (Fig. S4). On the typhoon-induced days in autumn, downdrafts
occurred over large areas above the PRD, especially above a height of ~4 km. Although updrafts can still be found near the
sea surface in this scenario, vertical wind speeds tended to be lower compared with those on the no-typhoon days in autumn,
which also suggests the enhancement of downdrafts caused by typhoons. In summer, the influence of downdrafts was found
over the PRD under 850 hPa on the typhoon-induced $O_3$ pollution days. However, overall, updrafts prevailed above the land
areas and downdrafts prevailed above the sea in both the typhoon-induced and no-typhoon scenarios in summer, which is
recognised as the structure of the East Asian summer monsoon cell (Chen et al., 1964; Jin et al., 2013; Ding et al., 2018). For
both updrafts and downdrafts, the absolute values of vertical wind speeds in the typhoon-induced scenario in summer were
overall higher than these in the corresponding no-typhoon scenario. Therefore, the approach of typhoons did not break the
structure of the summer monsoon cell, but rather they further strengthened the vertical motions above both land areas and sea.
These analyses suggest that typhoons do not necessarily lead to downdrafts during $O_3$ pollution periods in the PRD and its
adjacent areas; and in summer, vertical air motions affected by typhoons are more complicated than expected owing to the
existence of the East Asian summer monsoon.

We also explored the regions where downdrafts and updrafts occurred on a larger scale and their potential connections with
$O_3$ levels. As is shown in Fig. 5, though updrafts appeared in the PRD at 850 hPa on the typhoon-induced days in autumn,
downdrafts dominated in the region at 700 and 500 hPa. For the areas to the north of the PRD, the important role of downdrafts
was found at all three heights. In contrast to the no-typhoon days in autumn, downdrafts tended to cover much larger areas in
this scenario. Moreover, these areas at 850 and 700 hPa generally featured higher $O_3$ mixing ratios as well as lower RH (Fig.
S6) than others, which is a sign of possible direct downward $O_3$ transport (Roux et al., 2020; Wang et al., 2020). This part of
$O_3$ can notably aggravate near-ground $O_3$ pollution in the PRD. In contrast, in summer, updrafts dominated the PRD at various
heights in both scenarios. Besides the PRD, most of the regions near the coast were characterised by updrafts above the land
as well as downdrafts offshore, further indicating the ubiquity of the summer monsoon cell. By comparing the two scenarios
in summer, we found that typhoons resulted in more areas being influenced by updrafts. The areas with high $O_3$ levels did not
coincide with the downdraft-affected areas, and therefore, $O_3$ transported from the upper air may play a less significant role in
the typhoon-induced $O_3$ pollution in summer.

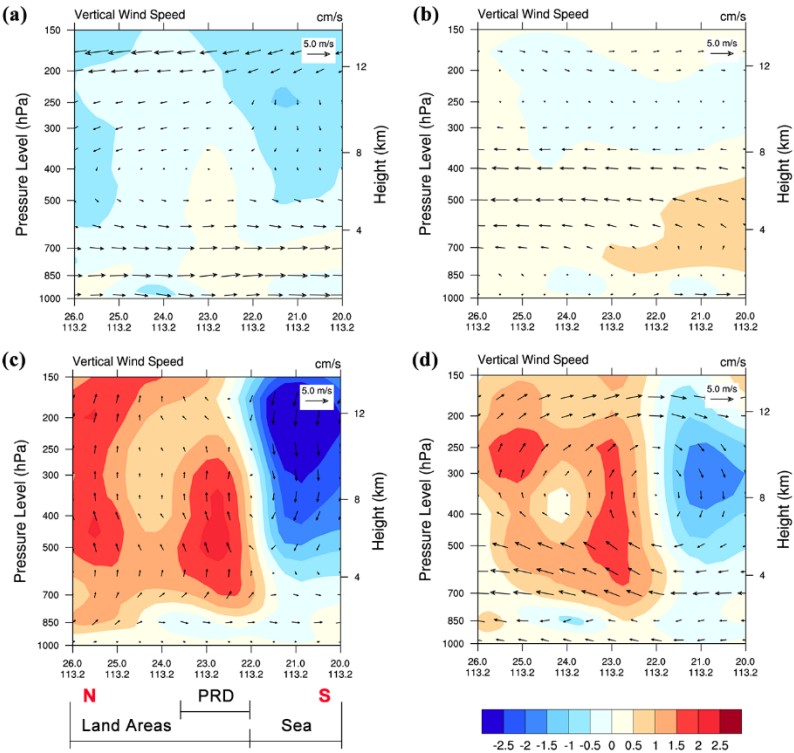


**Figure 4.** The cross sections of mean vertical wind field at 14:00 LT for the four scenarios: (a) autumn, typhoon-induced; (b) autumn, no-typhoon; (c) summer, typhoon-induced; and (d) summer, no-typhoon. Cross sections are from 26.0 °N to 20.0 °N along the 113.2 °E longitude line (Fig. S4). The vectors indicate meridional wind speed (m/s) and vertical wind speed (cm/s), and the contours indicate vertical wind speed (cm/s). PRD, the Pearl River Delta.

### 3.3 O₃ production conditions: comparison of clouds

Clouds efficiently reflect solar radiation (Liou, 1976), and therefore, they have a notable impact on the local formation of $O_3$. The comparison of cloud liquid water content in the cross section (Fig. 6, derived from the ERA-Interim datasets) suggests that typhoons generally resulted in fewer clouds in autumn but more clouds in summer, which agrees well with the comparison of cloud covers in Table 2. The presence of fewer clouds on the typhoon-induced days in autumn can be attributed to two reasons: the influence of dry air masses (indicated by lower RH in Table 2 and Fig. S6) and/or the hindrance of cloud formation by downdrafts. In summer, the strengthened updrafts above the land caused by typhoons favoured cloud formation, which is demonstrated by higher cloud liquid water content at the heights of 2–5 km and increases in medium and high cloud covers. In areas above the PRD below 850 hPa, downdrafts led to slight decrease of clouds in the typhoon-induced scenario in summer, which is also indicated by reduced low cloud cover. As a consequence of varied cloud covers in each scenario, on average, net surface solar radiation increased by 13% and decreased by 7% on the typhoon-induced days in autumn and summer, respectively (Table 2), which promoted and hindered $O_3$ production in the PRD during these two seasons, respectively.








**Figure 5.** $O_3$ mixing ratio (ppbV) and wind fields at the height of (a–d) 850 hPa, (e–h) 700 hPa, and (i–l) 500 hPa at 14:00 LT for the four scenarios: (a, e, i) autumn, typhoon-induced; (b, f, j) autumn, no-typhoon; (c, g, k) summer, typhoon-induced; and (d, h, l) summer, no-typhoon. The red triangle in each plot indicates the PRD. The gridded areas indicate that vertical wind speed is less than 0, or downdrafts occur.

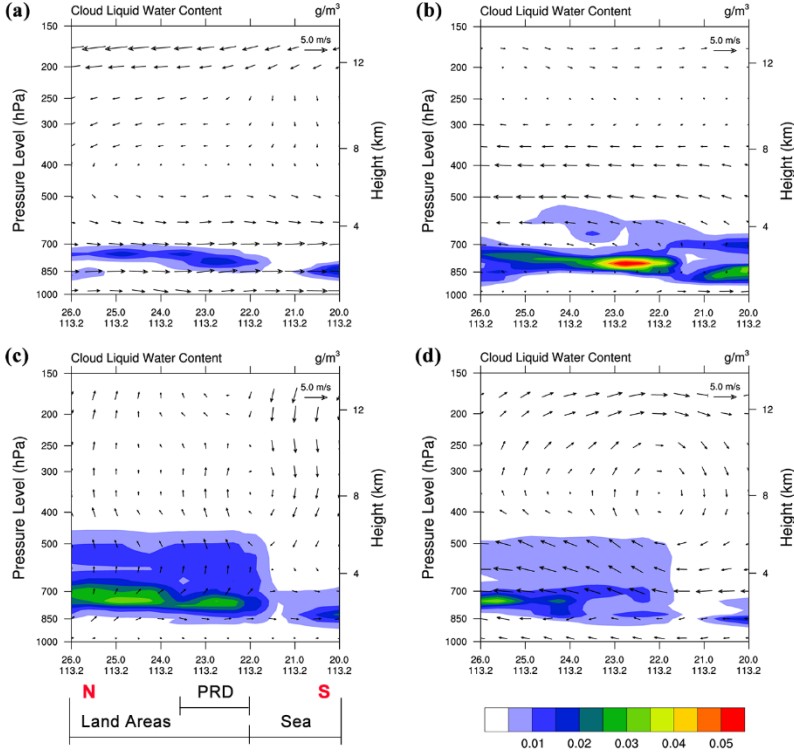

**Figure 6.** The cross sections of mean cloud liquid water content (g/m³) and wind vectors at 14:00 LT for the four scenarios: (a) autumn, typhoon-induced; (b) autumn, no-typhoon; (c) summer, typhoon-induced; and (d) summer, no-typhoon. Cross sections are from 26.0 °N to 20.0 °N along the 113.2 °E longitude line (Fig. S4). The vectors indicate meridional wind speed (m/s) and vertical wind speed (cm/s). PRD, the Pearl River Delta.

## 3.4 $O_3$ accumulation conditions: comparison of APRTs

The longer APRTs are, the more likely that $O_3$ produced by local emissions accumulates within the targeted region and notably contributes to near-ground $O_3$ pollution. In order to study the effect of typhoons on $O_3$ accumulation, we calculated APRTs in the PRD in the four scenarios (Fig. 7) for the further comparisons. On the typhoon-induced days in autumn, APRTs were typically 5–10 hours (mean = 9.5 hours) — shorter than those on the no-typhoon days in autumn (mean = 13.1 hours). In addition, lower APRT values occurred in the central part of the PRD, where high anthropogenic emissions of pollutants are distributed (Zheng et al., 2009). Despite more active $O_3$ chemistry discussed in the last section, locally sourced $O_3$ was less likely to accumulate within the PRD in this scenario, potentially limiting the contribution of local emissions for $O_3$. The comparison suggests opposite results in the summer scenarios, that is, APRTs on the typhoon-induced days (20–30 hours, mean = 21.0 hours) were overall higher than those on the no-typhoon days (15–25 hours, mean = 16.5 hours). This favoured the accumulation of locally sourced $O_3$ and offset the influence of weakened $O_3$ formation to some extent. In both seasons, typhoons did not cause more favourable conditions for $O_3$ production and accumulation simultaneously in the PRD, potentially





resulting in a less important role of local contributions in $O_3$ pollution here. More quantitative evaluations of the contributions
from multiple $O_3$ sources are discussed in Sect. 4.

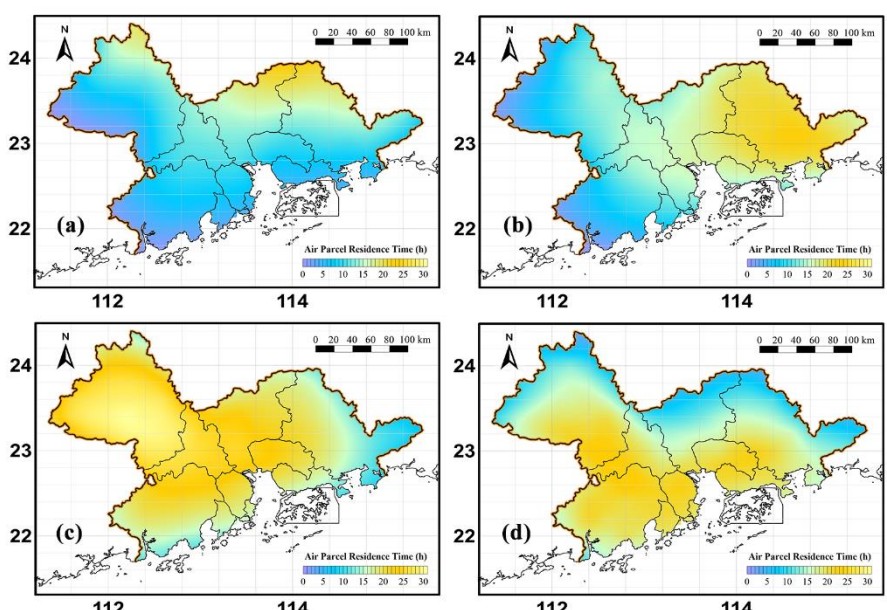


**Figure 7.** The spatial distributions of APRTs in the PRD for the four scenarios: (a) autumn, typhoon-induced; (b) autumn, no-typhoon; (c)
summer, typhoon-induced; and (d) summer, no-typhoon.
**3.5 Meteorological conditions on the close typhoon-induced days**
On the close typhoon-induced days in the two seasons, stronger north winds prevailed and total cloud cover was higher than
that on the no-typhoon days (Table 2), suggesting better conditions for the transport of $O_3$ but less favourable conditions for
$O_3$ production. As displayed in Fig. S7, the APRT values were significantly lower on the close typhoon-induced days (mean
= 6.6 hours, 12.9 hours in autumn and summer, respectively) than on the no-typhoon days, making it even harder for locally
sourced $O_3$ to accumulate within the PRD. Therefore, close typhoons are concluded to promote the transport of $O_3$ from the
outside and to reduce the contributions of $O_3$ produced from local emissions in a more notable way. In addition, close typhoons
led to stronger downdrafts in autumn and updrafts in summer than other scenarios in the same season (Fig. S8). It should be
noted that the structure of the summer monsoon cell near the PRD was destroyed in the close typhoon-induced scenario in
summer, indicating the stronger influence of typhoons on regional wind fields. The dominant role of $O_3$ transport during $O_3$
pollution days in this special scenario agrees well with the reported episode-based analyses (Lam et al., 2005; Li, 2013).





## 4 Comparisons of O₃ processes and sources

The comparisons of meteorological conditions served as qualitative evidence to determine the general influence of typhoons
on O₃ transport, production and accumulation in autumn and summer. Based on the comparison between the CMAQ modelling
results on typical O₃ pollution days in October 2015 and July 2016, more quantitative evidence can be presented. Figure 8
displays modelled mean O₃ MDA8 concentrations and wind fields (at 14:00 LT) in the four scenarios. Large standard-
exceedance (> 160 μg/m$^3$) areas were distributed in the PRD on most days, and the typhoon-induced days of both seasons
generally featured higher O₃ levels. The distinct wind fields for these scenarios, which were consistent with those in the longer
timespan (Fig. S5), indeed led to different spatial distributions of O₃. Generally, the most severe O₃ pollution occurred in the
downwind areas, such as the central and southern parts of the PRD on the typhoon-induced days in October 2015, the central
PRD on the no-typhoon days in October 2015, and the northern and eastern PRD on the no-typhoon days in July 2016. On the
typhoon-induced days in July 2016, high levels of O₃ accumulated around the PRE. In this section, we discuss the different
contributions of various O₃ processes and sources on these days to better understand the effect of typhoons on O₃ pollution in
the PRD.

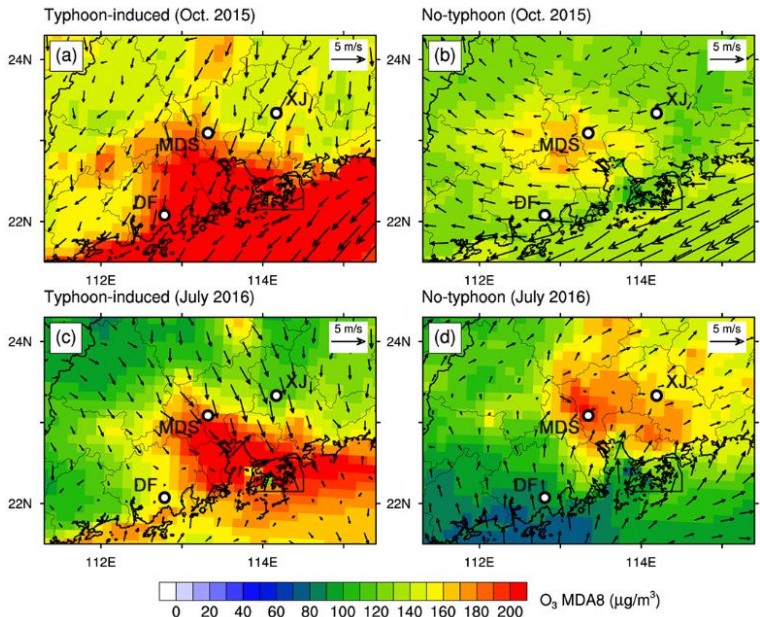

**Figure 8.** Modelling mean O₃ MDA8 concentrations (μg/m$^3$) and wind vectors (at 14:00 LT) on the representative O₃ pollution days: (a) the typhoon-induced days in October 2015 (14–16 and 21 October 2015); (b) the no-typhoon days in October 2015 (28 October and 3–5 November 2015); (c) the typhoon-induced days in July 2016 (7–8 and 30–31 July 2016); and (d) the no-typhoon days in July 2016 (22–26 and 29 July 2016). Three representative sites in the PRD are shown as black circles in the plots: XJ, Xijiao; MDS, Modiesha; DF, Duanfen.





### 4.1 O₃ processes: transport vs chemical process

The PA tool in CMAQ was used to quantify the contributions of transport and chemical process to the O$_3$ variations on O$_3$ pollution days under various scenarios in October 2015 and July 2016. As is shown in Fig. 9, the daytime (9:00–17:00 LT) O$_3$ PA results within the PRD in all scenarios share similar characteristics. Dry deposition dominated O$_3$ removal near the surface, and it also led to high gradients of O$_3$ concentrations that promote downward O$_3$ diffusion. Within the PBL (about 0–1 km in height), O$_3$ was mainly contributed by horizontal transport and chemical process, and vertical convection led to the drop of O$_3$ concentrations. However, differences existed between the O$_3$ PA results in four scenarios, indicating the impact of typhoons on the transport and production of O$_3$. In both months, typhoons led to notably higher contribution of horizontal transport to O$_3$, especially in the lower and middle part of the PBL. Within the PBL, on average, it increased from -0.9 ppb/h, -0.8 ppb/h to 1.2 ppb/h, 2.0 ppb/h under typhoon influence in autumn and summer, respectively. The comparison of the contribution of chemical process (in absolute rates) suggests that they had opposite effects in the two months — under typhoons, the contribution increased in October 2015 (from 4.0 ppb/h to 4.5 ppb/h within the PBL, or by 11.4%), but it decreased in July 2016 (from 7.1 ppb/h to 5.7 ppb/h within the PBL, or by -20.8%). In other words, typhoons promoted and hindered O$_3$ production in autumn and summer, respectively. These results agree well with the comparisons of O$_3$ transport and production conditions in the previous section.

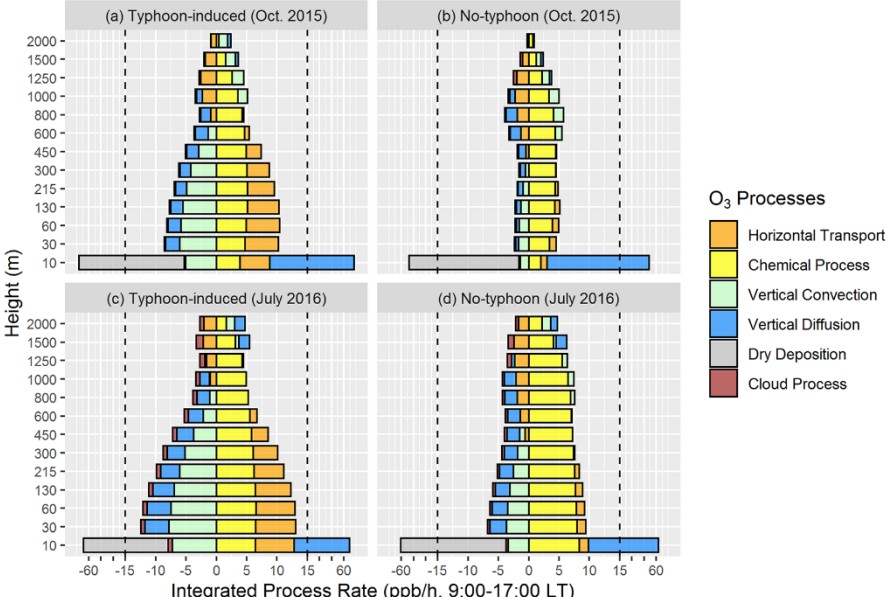

**Figure 9.** The daytime-mean (9:00–17:00 LT) hourly contributions of O$_3$ processes within the PRD in vertical layers 1–13 on representative O$_3$ pollution days: (a) the typhoon-induced days in October 2015 (14–16 and 21 October 2015); (b) the no-typhoon days in October 2015 (28 October and 3–5 November 2015); (c) the typhoon-induced days in July 2016 (7–8 and 30–31 July 2016); and (d) the no-typhoon days in July 2016 (22–26 and 29 July 2016).



### 4.2 O₃ sources: local sources vs regional sources

The contributions of various sources to $O_3$ within the PRD are determined by the combined impact of $O_3$ transport, production and accumulation. The results for the daytime (9:00–17:00 LT) $O_3$ SA near the ground (about 0–80 m in height) in four scenarios are illustrated in Fig. 10. For polluted regions within the PRD, stronger $O_3$ production under typhoons did not lead to a higher proportion of local contributions to $O_3$ pollution in October 2015 — it even decreased from 22% (on the no-typhoon days) to 17% (on the typhoon-induced days). The contributions of EC-China emissions and BCON, in contrast, increased slightly from 37%, 41% to 40%, 43%, respectively. The distinction of the $O_3$ SA results is more apparent for the summer scenarios, that is, typhoons resulted in growing contributions from $O_3$ transported from other regions (from 40% to 59%) but decreased local contributions (from 60% to 41%) in July 2016. More favourable $O_3$ accumulation conditions (indicated by higher APRTs on the representative typhoon-induced $O_3$ pollution days in summer (Fig. S9)) were far from sufficient to compensate for the effect of weakened $O_3$ production on the high contributions of local sources.

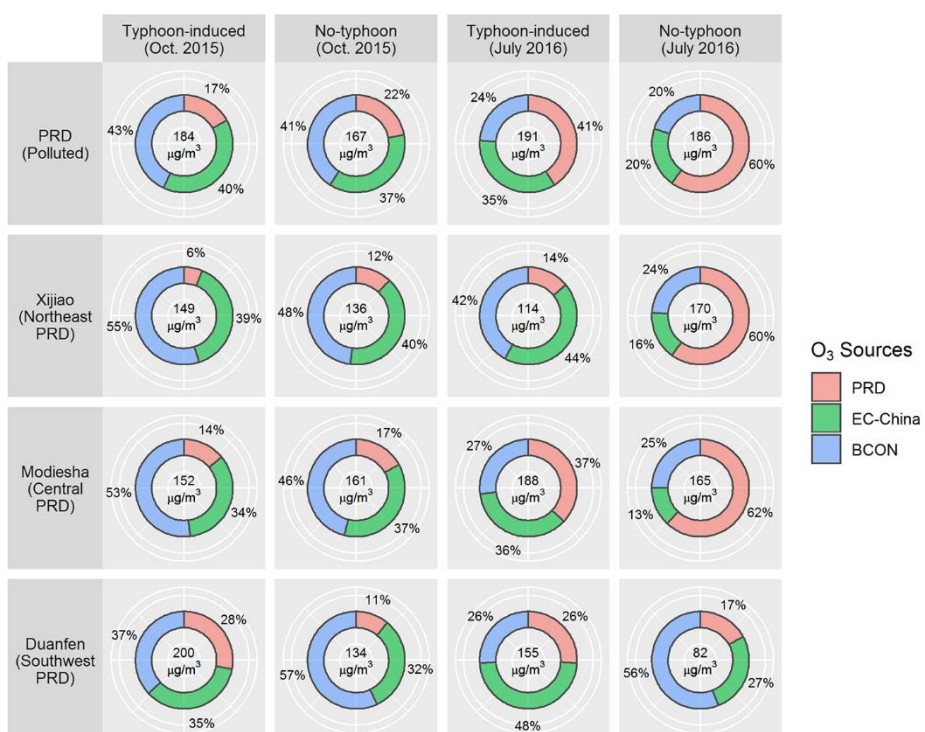

**Figure 10.** The $O_3$ SA near the ground (about 0–80 m in height) on representative $O_3$ pollution days for the four scenarios (the average results of 9:00–17:00 LT). The locations of the three representative sites (Xijiao, Modiesha and Duanfen) are shown in Fig. 8. PRD, the Pearl River Delta; EC-China, East China and Central China; BCON, the boundary conditions of the d02 modelling.

Furthermore, owing to the variations of wind fields, the comparison results of $O_3$ SA in different parts of the PRD may differ from the regional ones. For instance, while the comparisons of $O_3$ SA in the Xijiao and Modiesha site (located in the northeast



422 and central part of the PRD, respectively) agree well with those in the polluted regions of the PRD, higher contributions of

423 PRD emissions for $O_3$ can be found in the Duanfen site (located in the southwest part of the PRD) on the typhoon-induced

424 days of two months in comparison to these on the corresponding no-typhoon days (Fig. 10). Since the site was located in the

425 downwind region in the typhoon-induced scenario in October 2015 (Fig. 8a), enhanced $O_3$ production led by typhoons from

426 the massive emissions of $O_3$ precursors in the central PRD (Zheng et al., 2009) contributed to higher local contributions for $O_3$

427 pollution here (the highest local contribution in the PRD occurred in areas near the Duanfen site and almost reached 40% in

428 this scenario, which was even higher than that in the corresponding no-typhoon scenario (33%)). In the no-typhoon scenario

429 in July 2016, the site was located in the upwind regions under the prevailing of southwest winds, limiting the contributions of

430 local emissions for $O_3$ at the site (Fig. 8d). Thus, higher local contributions can also be found in the typhoon-induced scenario

431 in this month.

432 **5 Discussion and conclusions**

433 The significance of typhoons on $O_3$ pollution in the PRD calls for thorough evaluations of the different causes of $O_3$ pollution

434 with the appearance of typhoons in the Northwest Pacific. In this study, we revealed the different impacts of typhoons on $O_3$

435 transport, production and accumulation in the PRD (as summarised in Fig. 11) through systematic comparisons of

436 meteorological conditions, the contributions of various $O_3$ processes and sources in the typhoon-induced and no-typhoon

437 scenarios. We found that typhoons tended to promote $O_3$ transport towards the PRD, but failed to provide more favourable $O_3$

438 production and accumulation conditions simultaneously, which limited the contribution of local emissions to $O_3$ pollution.

439 Furthermore, there were also differences between the influence of typhoons on $O_3$ pollution in autumn and summer. More

440 favourable transport conditions occurred in the typhoon-induced scenario in autumn, which was characterised by higher wind

441 speeds and the increased influence of downdrafts. In summer, the mixed types of air masses in the typhoon-induced scenario

442 were likely to bring more $O_3$ into the PRD than the clean marine air masses in the no-typhoon scenario, also suggesting

443 enhanced $O_3$ transport under the influence of typhoons. Generally, typhoons led to cloudless conditions, stronger solar radiation,

444 and thus more rapid $O_3$ production in autumn, but shorter APRTs (5–10 hours) suggest that locally sourced $O_3$ was hard to

445 accumulate within the PRD. As a result, the contributions in percentage of local emissions to $O_3$ pollution decreased (slightly

446 by ~5% for the polluted regions of the PRD in October 2015). In contrast, in summer, intensified updrafts associated with

447 typhoons strengthened cloud formation, weakened solar radiation, and thus restrained local $O_3$ production. Longer APRTs (>

448 20 hour) under typhoon influence were far from sufficient to maintain high contributions of local emissions for $O_3$ pollution

449 (which decreased by ~20% for the polluted regions of the PRD in July 2016). However, due to the variations of wind fields

450 under different scenarios, the changes of local and transport contributions for $O_3$ led by typhoons were different in the

451 southwest part of the PRD, that is, higher contribution from emissions within the PRD and reduced transport contribution

452 occurred in the typhoon-induced scenarios in both seasons. As for the close typhoon-induced scenario, $O_3$ transport was further





strengthened, but meteorological conditions in the PRD became less favourable for both the production and accumulation of
$O_3$.

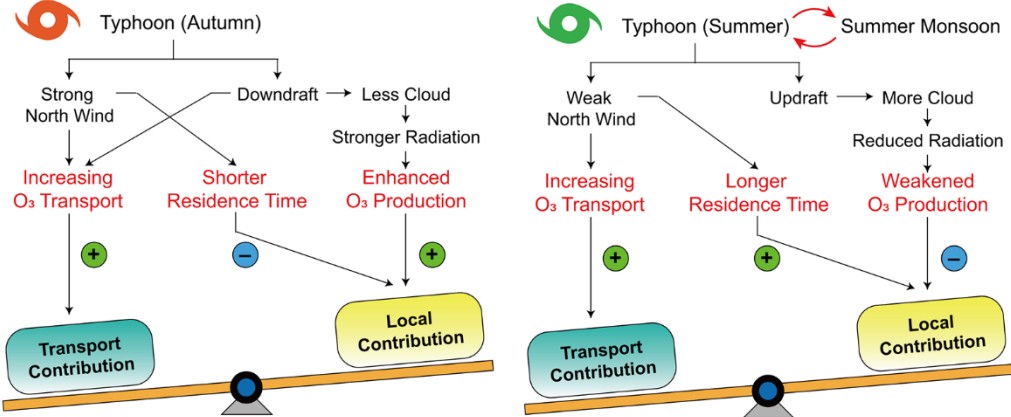

**Figure 11.** The summary of the causes of $O_3$ pollution in the PRD under typhoon influence in autumn and summer.
The East Asian monsoon, changing with seasons, has a pronounced impact on local meteorological conditions as well as the
characteristics of $O_3$ pollution in East China (He et al., 2008). The seasonal behaviour of the East Asian monsoon is likely to
result in the seasonally varied effect of typhoons on $O_3$ pollution in the PRD. In October, the summer monsoon has almost
finished its retraction and the winter monsoon is beginning (Ding, 1994). Thus, there are not many obstacles to the southward
movement of typhoon periphery and the transport of $O_3$ towards the PRD by the continental and coastal air masses. Large
downdraft-influenced areas in Central and South China occur in this scenario, and high $O_3$ levels and low RH in these areas
indicate the potentially important role of directly downward $O_3$ transport. In July, the summer monsoon reaches its strongest
(Ding, 1994). The interaction between typhoon periphery and the summer monsoon results in stagnation and enhanced updrafts
above the land areas of the PRD and its surroundings. Only when typhoon is close enough to the PRD is the stagnation
terminated and the structure of the summer monsoon cell broken. This also explains why some summertime typhoon-induced
$O_3$ episodes in the PRD can be typically divided into two periods, as stagnation leads to the accumulation of locally produced
$O_3$ in the first phase and strong northerly winds strengthen $O_3$ transport before the landing of typhoons (Lam et al., 2005; Li,
2013). It should be noted that updrafts, rather than downdrafts, prevailed on the typhoon-induced $O_3$ pollution days in summer.
High levels of $O_3$ did not necessarily occur in the regions dominated by downdrafts in this scenario, suggesting a less notable
connection between downdrafts and summertime $O_3$ pollution in the PRD. Further investigations are required to trace the
detailed process of downward $O_3$ transport, including the stratosphere-troposphere exchange (Stohl et al., 2003), in each
scenario, and quantify their contributions to near-ground $O_3$ pollution.





Some limitations remain in this study. We chose $O_3$ pollution days as individual samples, ignoring the influence of $O_3$ pollution
on the previous days. Thus, more detailed full-episode analyses are required. Moreover, owing to the small sampling size, the
influence of typhoons on $O_3$ pollution in the PRD is still not fully understood, including, for instance, the detailed connections
between the features of typhoons (intensity, position) and $O_3$ pollution. However, the comparisons of meteorological conditions,
$O_3$ processes and sources in different scenarios and seasons demonstrate the complex causes of typhoon-induced $O_3$ pollution
in the PRD — typhoons tend to enhance $O_3$ transport into the PRD in both seasons, but their impacts on the production and
accumulation of $O_3$ are completely different. As a result, emissions within (outside of) the PRD are likely to contribute less
(more) on the typhoon-induced $O_3$ pollution days than on the no-typhoon days, and more attention should be paid to controlling
anthropogenic emissions of $O_3$ precursors on a larger scale under typhoon influence. This study also suggests that a thorough
evaluation of $O_3$ transport, production and accumulation conditions can be applied to understand the causes of regional $O_3$
pollution not only in the PRD, but also in other regions.

*Data availability*. Data are available from the corresponding author upon request.

*Author contributions*. KQ, XW and YZ designed the study. KQ, XW, and TX did the simulation work, including the operation
of the WRF, SMOKE and CMAQ models. JS, HD, LZ and YZ provided observational results of field campaigns and the
routine monitoring datasets for the evaluation of model performance. KQ, XW, YY and YZ analysed the modelling results.
KQ, XW, YY and YZ wrote and revised this paper, with critical feedbacks from all other authors.

*Competing interests*. The authors declare no conflict of interest.

*Acknowledgements*. This work was supported by the National Key Research and Development Program of China (Grant No.
2018YFC0213204,2018YFC0213506) and the National Science and Technology Pillar Program of China (Grant No.
2014BAC21B01).



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
