# Peer review of "A Comparative Study to Reveal the Influence of Typhoons on the"

_Atmospheric Chemistry and Physics, 2020_

## Author Comment (AC1)

**Response to Referee #2**

**Comment:**
The manuscript provides thorough analysis of the influence of typhoons on the occurrence of ozone episodes in the Pearl River Delta, China.

Despite many papers (correctly referenced by the Authors) have been published concerning ozone pollution in the area, the manuscript resumes the different aspect of ozone episodes development and can be a guide through previous literature. The modelling section is, from my point of view, the most important to provide a clear support to the hypothesis and correlations provided by the previous analysis. A possible missing point is the evaluation of the overall import/export of ozone, to estimate if the PRD region is responsible of a net export of ozone increasing the amount of pollutant over the region.

The overall result that typhoon influenced $O_3$ episodes have a major contribution from long range transport (from outside the model domain) and advection from nearby China regions has relevant policy implications that are only quickly commented in the conclusions and would merit a wider discussion. Local $O_3$ precursor emission control can be expected to have a limited effectiveness and regional policies seem definitely needed, at China national level but even at South and East Asia regional level, to be able to reduce population exposure and overall ozone production.

The manuscript is well written and needs few clarifications/integrations to reach publication quality.

**Response:**
Thanks for your positive comments and valuable suggestions to help us improve the manuscript. The responses to the comments (in blue) and corresponding revisions (in red) are presented as follows.

**Comment:**
1. (Lines 40-42, Introduction) This is due to the $O_3$ persistence in the atmosphere due to its relatively long lifetime in the atmosphere. The Authors should consider mentioning it.

**Response**:
According to this suggestion, we added this content:
"Due to the relatively long lifetime of $O_3$ (~22 days; Stevenson et al., 2006), it can accumulate locally, or be transported to downwind regions."

**Comment:**
2. (Line 49, Introduction) "of" just before the symbol ">" should be removed.

**Response**:
We deleted "of" before ">":
"… seven out of the nine most severe $O_3$ episodes (regional-mean maximum 8-h average $O_3$ concentrations > 240 μg/m$^3$) …"

**Comment:**
3. (Lines 104-105, Method) Why precipitation is not considered?

**Response**:

The main focus of this study is the comparison between typhoon-induced and no-typhoon $O_3$ pollution. As shown in Table R1, sunny, cloudy or overcast weathers can be found on all $O_3$ pollution days in autumn and over 60% of $O_3$ pollution days in summer, when the precipitation was namely 0. Therefore, the comparison of precipitation does not help with the comparisons in this study.

**Table R1**    The numbers (percentages) of $O_3$ pollution days corresponding to different weathers in Guangzhou in four scenarios (data source: tianqihoubao (historical weather records), http://www.tianqihoubao.com/lishi/guangzhou.html)

| Weathers | Oct. 2014–2018 Typhoon-induced | Oct. 2014–2018 No-typhoon | July 2014–2018 Typhoon-induced | July 2014–2018 No-typhoon |
|---|---|---|---|---|
| Sunny | 31 (63%) | 12 (41%) | 13 (29%) | 2 (20%) |
| Cloudy | 16 (33%) | 17 (59%) | 18 (40%) | 4 (40%) |
| Overcast | 2 (4%) | / | / | / |
| Shower | / | / | / | 1 (2%) |
| Thunder-shower | / | / | 12 (27%) | 3 (30%) |
| Light/Moderate Rain | / | / | 1 (2%) | / |
| Heavy Rain | / | / | 1 (2%) | / |

We added this information in line 230 of the ACPD manuscript:

"The parameters from … were used in the comparison (since all $O_3$ pollution days in October and over 60% of $O_3$ pollution days in July were characterized with sunny, cloudy, or overcast weathers with no rainfall in the PRD (Table S4, represented by the weather in Guangzhou), precipitation was not considered in the comparisons)."

And also, Table R1 was added to the Supplement as Table S4.

**Comment:**

4. (Lines 122-123, Method) Does the mentioned "30%" refers to the total number of days or to the number of polluted days?

**Response**:

"30%" refers to the total number of days. To be clearer, we revised the sentence into:

"Although there were more $O_3$ pollution days in October than in July, $O_3$ pollution under typhoon influence occurred on ~30% days of both months."

**Comment:**

5. (Lines 123-125, Method) This consideration seems convincing for July only. For October the difference of values with/without typhoons seems rather small.

**Response**:

That is correct. To express it more precisely, we revised the sentence into:

"Higher $O_3$ MDA1 and MDA8 values can be found with the appearance of typhoons in comparison with days without typhoons in July, whereas these values are similar in October, indicating the important role of typhoons in $O_3$ pollution in the PRD."

**Comment:**

6. (Lines 129-130, Method) This is not clear, what is the reason to discard 5 episodes?

**Response**:

These five $O_3$ pollution days were affected by typhoons located to the due north or southwest of the PRD, while the remaining days in October and July, which are the majority of the samples, were affected by typhoons located to the east of the PRD. Since the influence of typhoon on wind field and other meteorological parameters might be associated with the direction of typhoon to the PRD, we discarded these five days featuring with different typhoon directions to minimize the disturbance of typhoon locations in the comparisons. In order to explain more clearly, we revised the sentence into: "As is shown in Fig. 1, all $O_3$ pollution days in October and most $O_3$ pollution days in July under typhoon influence were associated with typhoons to the east of the PRD, which were more likely to cause $O_3$ pollution (Chow et al., 2018). In order to minimize the disturbance of typhoon directions in the comparisons, we removed the remaining five $O_3$ pollution days in July with typhoons located to the due north or southwest of the PRD from the analyses."

**Comment:**

7. (Line 157, Method) Why the endpoint of the back trajectories has been set to 500 m and not nearer the surface?

**Response**:

The height of 500 m is near the middle of the planetary boundary layer (PBL) during the daytime (Guo et al., 2016). Backward trajectories arriving at the height of 500 m can well represent the effect of long-range transport on near-ground $O_3$ pollution (Park et al., 2007). By contrast, backward trajectories arriving at the surface are under the notable disturbance of the surface (including buildings, plants and other objects). To give more details, we revised the sentence into:
"The Modiesha site (23.1°N, 113.3°E; Fig. S1b), which is located in the central part of the PRD, was the endpoint of backward trajectories. Its height was set as 500 m above the ground to better represent the effect of long-range transport on $O_3$ pollution, as well as to minimize the disturbance of objects near the surface to the transport (Park et al., 2007)."

**Comment:**

8. (Line 161, Method) What is the horizontal space resolution of the mentioned matrix?

**Response**:

The horizontal space resolution of the matrix is 0.2°×0.2°. We added this information:
"To calculate APRTs in the PRD, we designed a 21×15 point matrix (resolution: 0.2°×0.2°) that embraces the whole PRD (Fig. S4)…"

**Comment:**

9. (Line 163, Method) Setting the trajectory starting points to 100 m seems reasonable for industrial emissions, but it seems high for road transport related emissions or other surface sources.

**Response**:

Firstly, 100 m was selected to represent the height of all local emissions. For instance, according to

the local emission inventory used in this study, power stations (contributing to ~32% of $NO_x$ emissions in the PRD) emit pollutants at the height of 200–500 m, industrial sources (contributing to ~12% of $NO_x$ emissions and ~45% of VOCs emissions in the PRD) emit pollutants at the height of 0–200 m, and road transport sources (contributing to ~54% of $NO_x$ emissions and ~31% of VOCs emissions in the PRD) emit pollutant near the ground. Therefore, 100 m is close to the mean height of all emissions in the PRD.

Secondly, the focus of this study is the influence of weather conditions (typhoons) on $O_3$ processes. However, local topography and objects near the surface could also lead to different characteristics of trajectories. 100 m, instead of lower height, was also chosen to reduce the disturbance of the surface. Based on these reasons, we revised the sentence to make it clearer:

"The height of all points was set as 100 m above the ground to represent the height of all local emissions and to reduce the disturbance of the surface, as well."

**Comment:**

10. (Line 165-167, Method) Were time durations attributed to points and then gridded? on which target grid?

**Response**:

Yes. APRT was attributed to the starting point of the trajectory, and the mean APRTs in all points were interpolated using the Kriging method to obtain field results. We revised the sentence into:

"Afterwards, the length of time each trajectory remained within the administration borders of the PRD, i.e., APRT, was calculated and attributed to its starting point. APRTs in each point were averaged, and these averaged APRT values in all points were interpolated using the Kriging method to obtain field results for the further comparisons."

**Comment:**

11. (Line 168, Method) The sentence "model, the CMAQ model (version 5.0.2)" should be simplified to avoid useless word repetition.

**Response**:

Accepted. We simplified the sentence into:

"We utilised the widely used 3D chemical transport model CMAQ (version 5.0.2) to investigate …"

**Comment:**

12. (Line 173, Method) The meaning of the sentence "all $O_3$ pollution days in these two months served as representative $O_3$ pollution days under multiple scenarios." is not clear.

**Response**:

To make it clearer, we revised the sentence into:

"… all typhoon-induced and no-typhoon $O_3$ pollution days in these two months served as representative $O_3$ pollution days in the comparisons."

**Comment:**

13. (Line 176-178, Method) The meaning of this sentence is not clear.

**Response**:

To make it clearer, we revised the sentence into:

"The results of daytime (9:00–17:00 LT) $O_3$ PA and SA on representative $O_3$ pollution days were averaged for the typhoon-induced and no-typhoon scenarios in autumn (October 2015) and summer (July 2016) and used in the comparisons."

**Comment:**

14. (Line 180, Method) "CMAQ model", or "model application" would read better that "CMAQ modelling".

**Response**:

Accepted. We revised the sentence into:

"The main setups of the CMAQ model are presented as follows."

**Comment:**

15. (Line 210-214, Method) Alternatively to zero emission a fractional reduction could be applied to reduce non linearity. See e.g. https://fairmode.jrc.ec.europa.eu/activity/ct1

The Authors should comment this alternative approach and the reasons supporting their choice.

**Response**:

In order to make the comparisons between $O_3$ sources in the typhoon-induced and no-typhoon scenarios, we need to know the sources of all $O_3$ in the region, and the results need to be unambiguous and additive but may not be dynamic. Therefore, we chose the Brute Force Method with the allocation of the non-linear contributions between emissions from the PRD and EC-China.

However, the dynamicity of source apportionment (SA) method is important for policy-making. According to the report by Thunis et al. (2020) and also the review of Thunis et al. (2019), SA using the fractional reduction method is unambiguous, dynamic, and additive, thus is suitable to evaluate the effects of emission reductions. In future, it is still required to further evaluate the efficiency of local and non-local emission reduction for alleviating $O_3$ pollution in the PRD in different scenarios based on this method.

We added this information in line 207 of the ACPD manuscript:

"In order to identify the sources of all $O_3$ in the PRD, we used the classic Brute Force Method …"

in lines 214 of the ACPD manuscript:

"…, biases may occur between the results of two types of BFM methods, leading to the non-additivity of the results (Clappier et al., 2017)."

and in the *Discussion and conclusion* part (line 484 of the ACPD manuscript) as well:

"…under typhoon influence. For air quality management, it is suggested to comprehensively evaluate the efficiency of fractional local and non-local emission reductions to reduce $O_3$ levels in the PRD in different scenarios (Thunis et al., 2019; Thunis et al., 2020)."

**Comment:**

16. (Lines 227-231, Comparison of meteorological conditions) It is not clear how ERA-Interim fields have been processed. Has a gridded area been processed or timeseries have been extracted in few points? What choice has been done and why?

**Response**:

In order to be consistent with the comparison using the surface meteorological monitoring dataset,

we extracted meteorological parameters at 29 national meteorological sites within the PRD from the ERA-Interim re-analysis dataset in the comparisons. We added this information in the sentence:

"The parameters from routine monitoring datasets (including air temperature, RH, wind speed, zonal and meridional wind speeds measured at 14:00 LT of all $O_3$ pollution days at 29 national meteorological sites within the PRD (Fig. S1a)) and the ERA-Interim re-analysis (including all near-surface parameters from the analysis and forecast fields introduced in Sect. 2.1, extracted at the same time and the locations of sites as these in routine monitoring datasets) were used in the comparison…"

**Comment:**

17. (Lines 228-229, Comparison of meteorological conditions) The reference to first and second categories in Sect 2.1 is not clear.

**Response**:

"The parameters of the first and second categories in Sect. 2.1" indicate "near-surface parameters from the analysis fields" and "near-surface parameters from the forecast fields" introduced in Sect. 2.1. We revised the sentence to make it clear:

"The parameters from routine monitoring datasets (including air temperature, RH, wind speed, zonal and meridional wind speeds measured at 14:00 LT of all $O_3$ pollution days at 29 national meteorological sites within the PRD (Fig. S1a)) and the ERA-Interim re-analysis (including all near-surface parameters from the analysis and forecast fields introduced in Sect. 2.1, extracted at the same time and the locations of sites as these in routine monitoring datasets) were used in the comparison…"

**Comment:**

18. (Lines 269-271, Comparison of meteorological conditions) This sentence is not clear. Does it mean that during the summer the air masses advected by slow wind are expected to bring higher $O_3$ concentration? Or with low wind speed local phenomena would prevail on advection?

**Response**:

As mentioned in the manuscript, "the higher wind speeds and/or $O_3$ levels in the transported air masses are, the more likely $O_3$ transport plays an increasingly important role in $O_3$ pollution." Since the comparisons using the Mann-Whitney U test suggest that there was no statistically significant difference between wind speeds in the typhoon-induced and no-typhoon scenario in summer, as shown in Table 2, the $O_3$ level became the factor to determine if more $O_3$ was transported to the PRD, which was generally associated with the types of air masses influencing the PRD (higher $O_3$ levels for the continental and coastal air masses, and lower $O_3$ level for the marine air mass; Zheng et al., 2010). Therefore, in the typhoon-induced scenario in summer, the increasing influence of continental and coastal air masses (or "more polluted air masses") ensured that more $O_3$ is transported to the PRD. The discussion about the comparison of wind speed can be a distraction here, so we deleted the relative content and revised the sentence into:

"The increasing influence of much more polluted air masses (continental and coastal air masses) led by typhoon ensured that more $O_3$ was transported to the PRD, thus typhoons also tended to increase the contribution of transport to $O_3$ pollution in the PRD in summer."

**Comment:**

19. (Lines 279-280, Comparison of meteorological conditions) The reference to Sect 2.1 is not clear.

**Response**:
"the parameters of the third category in Sect. 2.1" indicates "upper air parameters at multiple heights" introduced in Sect. 2.1. We revised the sentence into:
"… the ERA-Interim reanalysis dataset (including all upper air parameters at multiple heights introduced in Sect. 2.1) …"

**Comment:**
20. (Lines 280-281, Comparison of meteorological conditions) Are values in Figure 4 mean values over the considered time period?

**Response**:
Yes. Figure 4 shows the distributions of mean vertical wind speed at 14:00 LT of all $O_3$ pollution days corresponding to the typhoon-induced and no-typhoon scenarios of two seasons. We added this information in lines 301–302:
"The contours in Fig. 4 show the cross sections of mean vertical wind speeds at 14:00 LT of all $O_3$ pollution days corresponding to the typhoon-induced and no-typhoon scenarios of two seasons, which were made along the 113.2°E longitude line, from 26.0°N to 20.0°N (Fig. S4)."

**Comment:**
21. (Lines 282, Comparison of meteorological conditions) Downdrafts seem to be at higher levels from the Figure. Please refer to the Figure vertical scale in hPa to be better understood by the reader.

**Response**:
We revised the sentence into:
"…downdrafts occurred over large areas above the PRD, especially above a height of ~700 hPa."
We also revised the sentence in line 319 of the ACPD manuscript into:
"…which is demonstrated by higher CLWC at the heights of 500–850 hPa …"

**Comment:**
22. (Lines 314, Comparison of meteorological conditions) How values in Figure 6 have been computed from ERA-Interim fields? Are they mean values?

**Response**:
Yes. Figure 6 shows the distributions of mean cloud liquid water content at 14:00 LT of all $O_3$ pollution days corresponding to the typhoon-induced and no-typhoon scenarios of two seasons. We added this information in the sentence:
"Figure 6 displays the cross sections of mean ERA-Interim cloud liquid water contents (CLWC) at 14:00 LT of all $O_3$ pollution days corresponding to the typhoon-induced and no-typhoon scenarios of two seasons, which were also made along the 113.2°E longitude line, from 26.0°N to 20.0°N (Fig. S4). The comparison of CLWC in the cross sections suggests …"

**Comment:**
23. (Lines 343-344, Comparison of meteorological conditions) The meaning of the sentence is obscure, what does "and offset the influence of weakened $O_3$ formation to some extent." mean?

**Response**:

Weakened $O_3$ formation led to lower local contributions to $O_3$ pollution, while the accumulation of locally sourced $O_3$ led to higher local contributions to $O_3$ pollution. We revised the sentence into: "This favoured the accumulation of locally sourced $O_3$, and, to some extent, offset the influence of weakened $O_3$ formation to maintain high contributions of local emissions to $O_3$ pollution."

**Comment:**
24. (Lines 344-346, Comparison of meteorological conditions) Please relate to the unfavourable / favourable conditions for ozone formation shown in the previous sections.

**Response:**
According to this suggestion, we revised the sentence into:
"Based on the comparison of $O_3$ production conditions in the previous section and the comparison of $O_3$ accumulation conditions in this section, typhoons did not provide more favourable conditions for $O_3$ production and accumulation simultaneously in the PRD in both autumn and summer, thus potentially resulting in a less important role of local contributions in $O_3$ pollution here."

**Comment:**
25. (Lines 427-428, Comparisons of $O_3$ processes and sources) I can't find this number in Figure 10.

**Response:**
The number is not shown in this original manuscript or supplement. We added this information as Fig. S10 in the Supplement:

[Figure]

**Figure S10.** The average local contributions (in percentage, %) to daytime (9:00–17:00 LT) $O_3$ and wind vectors (at 14:00 LT) on the representative $O_3$ pollution days: (a) the typhoon-induced days in October 2015 (14–16 and 21 October 2015); (b) the no-typhoon days in October 2015 (28 October and 3–5 November 2015); (c) the typhoon-induced days in July 2016 (7–8 and 30–31 July 2016); and (d) the no-typhoon days in July 2016 (22–26 and 29 July 2016). Three representative sites in the PRD are shown as black circles in the plots: XJ, Xijiao; MDS, Modiesha; DF, Duanfen.

and also mention it before the sentence:
"…(as the distribution of local contributions in percentage to daytime $O_3$ shown in Fig. S10, the

highest local contribution in the PRD occurred in areas near the Duanfen site and …).”

**Comment:**

26. (Lines 482-483, Discussion and conclusions) This discussion about anthropogenic emissions control is relevant and should be expanded to provide useful input to air quality management and suggestions to conceive measures capable to reduce the population exposure and the production of ozone at global scale.

**Response**:

We expanded our discussions into:

“… As a result, emissions within (outside of) the PRD are likely to contribute less (more) on the typhoon-induced $O_3$ pollution days than on the no-typhoon days. In order to effectively alleviate $O_3$ pollution and to reduce the population exposure in the PRD, more attention should be paid to controlling anthropogenic emissions of $O_3$ precursors on a larger scale, rather than focusing on local emissions, under typhoon influence. For air quality management, it is suggested to comprehensively evaluate the efficiency of fractional local and non-local emission reductions to reduce $O_3$ levels in the PRD in different scenarios (Thunis et al., 2019; Thunis et al., 2020). This study also suggests that a thorough evaluation of $O_3$ transport, production and accumulation conditions can be applied to understand the causes of regional $O_3$ pollution not only in the PRD, but also in other regions. The results will help find efficient strategies to alleviate regional $O_3$ pollution as well as to reduce its adverse effects.”

**Reference**

Chow, E. C., Li, R. C., and Zhou, W.: Influence of tropical cyclones on Hong Kong air quality, Adv. Atmos. Sci., 35(9), 1177–1188, https://doi.org/10.1007/s00376-018-7225-4, 2018.

Clappier, A., Belis, C. A., Pernigotti, D., and Thunis, P.: Source apportionment and sensitivity analysis: two methodologies with two different purposes, Geosci. Model Dev., 10, 4245–4256, https://doi.org/10.5194/gmd-10-4245-2017, 2017.

Guo, J., Miao, Y., Zhang, Y., Liu, H., Li, Z., Zhang, W., He, J., Lou, M., Yan, Y., Bian, L., and Zhai, P.: The climatology of planetary boundary layer height in China derived from radiosonde and reanalysis data, Atmos. Chem. Phys., 16, 13309–13319, https://doi.org/10.5194/acp-16-13309-2016, 2016.

Park, S. K., O'Neill, M. S., Stunder, B. J., Vokonas, P. S., Sparrow, D., Koutrakis, P., and Schwartz, J.: Source location of air pollution and cardiac autonomic function: trajectory cluster analysis for exposure assessment, J. Expo. Sci. Env. Epid., 17(5), 488-497, https://doi.org/10.1038/sj.jes.7500552, 2007.

Stevenson, D. S., Dentener, F. J., Schultz, M. G., Ellingsen, K., van Noije, T. P. C., Wild, O., Zeng, G., Amann, M., therton, C. S., Bell, N., Bergmann, D. J., Bey, I., Butler, T., Cofala, J., Collins, W. J., Derwent, R. G., Doherty, R. M., Drevet, J., Eskes, H. J., Fiore, A. M., Gauss, M., Hauglustaine, D. A., Horowitz, L. W., Isaksen, I. S. A., Krol, M. C., Lamarque, J.-F., Lawrence, M. G., Montanaro, V., Müller, J.-F., Pitari, G., Prather, M. J., Pyle, J. A., Rast, S., Rodriguez, J. M., Sanderson, M. G., Savage, N. H., Shindell, D. T., Strahan, S. E., Sudo, K., and Szopa, S.: Multimodel ensemble simulations of present-day and near-future tropospheric ozone, J. Geophys. Res., 111, D08301, https://doi.org/10.1029/2005JD006338, 2006.

Thunis, P., Clappier, A., Pirovano, G.: Source apportionment to support air quality management practices, A fitness-for-purpose guide (V 3.1), EUR30263, Publications Office of the European Union, ISBN 978-92-76-19744-7, doi:10.2760/47145, JRC120764, 2020.

Thunis, P., Clappier, A., Tarrason, L., Cuvelier, C., Monteiro, A., Pisoni, E., Wesseling, J., Belis, C. A., Pirovano, G., Janssen, S., Guerreiro, C., and Peduzzi, E.: Source apportionment to support air quality planning: Strengths

and weaknesses of existing approaches, Environ. Int., 130, 104825, https://doi.org/10.1016/j.envint.2019.05.019, 2019.

---

## Author Comment (AC2)

**Response to Referee #1**

**Comment:**

This work is of heavy workload and detailed analysis. Effects of typhoon on the transport, production, accumulation of $O_3$ are presented. Long time series of observations make the conclusions convinced. A series of sensitivity experiments are conducted to help understand how the differing location of typhoon would influence $O_3$ pollution in the PRD.

**Response:**

Thanks for your positive comments and valuable suggestions to help us improve the manuscript. The responses to the comments (in blue) and corresponding revisions (in red) are presented as follows.

**Comment:**

1. As mentioned in line 127, the differing location of typhoon will have diverse effects on $O_3$ pollution. In term of relationship between typhoon location and $O_3$ pollution, in what condition will the transport dominate, and in what condition will the accumulation lead? Likewise, how typhoon location affects the promotion/reduction of $O_3$ production? It would be better to summarize the general rule if possible, and show it in the conclusions.

**Response**:

Thanks for this good suggestion. Chow et al (2018) analyzed the connections between typhoon locations and $O_3$ levels in Hong Kong based on 2000–2015 $O_3$ observation data, and indicated that typhoons located to the east (south) of Hong Kong tend to cause more (less) severe $O_3$ pollution. However, according to our knowledge, there is still no report about how different typhoon locations lead to varied $O_3$ levels and processes (transport, production and accumulation) in the whole PRD or adjacent regions.

In this study, we focus on the overall differences of $O_3$ transport, production and accumulation with and without the influence of typhoons in the PRD. The comparison also suggests varied effects of typhoon on the above $O_3$ processes in autumn and summer. These conclusions can provide useful suggestions for efficient $O_3$ reduction in the PRD. However, the detailed connections between typhoon locations and $O_3$ processes were not involved in this study. We wish to explore this important question in the next-step studies based on the collections of longer-term $O_3$ observation datasets in the PRD.

Similar to this suggestion, the following content was presented in the *Discussion and conclusions* part of the manuscript, in lines 477–479 of the ACPD manuscript:

"Moreover, owing to the small sampling size, the influence of typhoons on $O_3$ pollution in the PRD is still not fully understood, including, for instance, the detailed connections between the features of typhoons (intensity, position) and $O_3$ pollution."

**Comment:**

2. In line 305-306, authors declare that vertical transport plays less significant role in the typhoon-induced $O_3$ pollution in summer, however, as what has been shown in figure 9, vertical transport contributes significantly in $O_3$ production. It makes me confused. Please give the explanation.

**Response**:

In lines 305-306 of the ACPD manuscript, "vertical transport" indicates the influence of large-scale air motion, namely, downdraft or updraft, which was mainly found at the height of above ~850 hPa. In Fig. 9, "vertical transport" includes vertical convection and diffusion (including diffusion caused by $O_3$ gradients and turbulent mixing within the planetary boundary layer (PBL; about 0-1 km in height, Guo et al., 2016)), which is different from the former one. According to the Process Analysis results (Fig. 9), dry deposition led to rapid $O_3$ removal near the surface, as well as high gradients of $O_3$ concentrations that promote downward $O_3$ diffusion. Therefore, vertical transport contributes significantly to $O_3$ in the first layer. But in other layers within the PBL, both vertical convection and diffusion served as sink processes for $O_3$.

Besides specifying the "vertical transport" in Fig. 9 as "vertical convection" and "vertical diffusion" in the last revision, the above discussions on Fig. 9 were also presented in the manuscript, in lines 386–389:
"Dry deposition dominated $O_3$ removal near the surface, and it also led to high gradients of $O_3$ concentrations that promote downward $O_3$ diffusion. Within the PBL (about 0–1 km in height), $O_3$ was mainly contributed by horizontal transport and chemical process, and vertical convection led to the drop of $O_3$ concentrations."

**Reference**

Chow, E. C., Li, R. C., and Zhou, W.: Influence of tropical cyclones on Hong Kong air quality, Adv. Atmos. Sci., 35(9), 1177–1188, https://doi.org/10.1007/s00376-018-7225-4, 2018.

Guo, J., Miao, Y., Zhang, Y., Liu, H., Li, Z., Zhang, W., He, J., Lou, M., Yan, Y., Bian, L., and Zhai, P.: The climatology of planetary boundary layer height in China derived from radiosonde and reanalysis data, Atmos. Chem. Phys., 16, 13309–13319, https://doi.org/10.5194/acp-16-13309-2016, 2016.

---

## Referee Report (RR1)

This manuscript systematically studied the difference between the typhoon-induced and no-typhoon O3 pollution in the PRD area of China. It elucidated the influence of typhoon on O3 transport, production and accumulation and found the seasonal difference of such influence. The revised version of the manuscript would be a good fit to ACP and publishable if minor comments below are addressed.

(1) Line 91: "We mainly used the ERA-Interim re-analysis product in the analyses due to its more available parameters and high spatial coverage". Regarding "more available parameters", does it compare to observations or to other reanalysis datasets such as GDAS? Why "mention spatial coverage" here if only ERA-Interim extractions at the monitoring locations are used.

(2) Line 97:"near-surface parameters from the forecast fields". What does it mean "forecast fields" if these fields came from the ERA-Interim re-analysis?

(3) Line 124: "Higher O3 MDA1 and MDA8 values can be generally found with the appearance of typhoons in comparison with days without typhoons in July, whereas these values are similar in October, further indicating the important role of typhoons in O3 pollution in the PRD." Better to just delete ", whereas these values are similar in October", and replace it with "and October".

(4) Line 158: "Hysplit model (Stein et al., 2015) with the Global Data Assimilation System (GDAS) and Line 190: "The Weather Research and Forecasting (WRF) model (version 3.2) provided the meteorological fields used as inputs." Why not use WRF outputs in Hysplit for consistency? Was GDAS or ERA-Interim used to drive WRF? What's the reason to use two different reanalysis datasets in the same study, considering the potential inconsistency between them?

(5) Line 191: "… process the anthropogenic and biogenic emission files". "Emission files" better to be "emissions".

(6) Line 219: "… the difference between two sensitivity cases where emissions expect Ei and all of the emissions are zeroed out, respectively". Is "expect" in fact "except"?

(7) Line 303: "scenario in summer were overall higher than these in the corresponding no-typhoon scenario". Better to replace "these" with "those"?

(8) Supplement Figure S11: "Comparisons between the observational and modelling mean O3 MDA8, daily NO2 and NMHCs concentrations in the PRD." Better to use "observed and modeled", instead of "observational and modelling".

(9) Figure S11 and Table S5. The conclusions of this study that based on modeling results rely on accuracy of the simulations, especially on consistency in performance of the simulations of different seasons as well as of different scenarios that with or without typhoon influence. Is there any significant performance difference between typhoon induced and no-typhoon scenarios for simulated meteorology and air quality?